# Focusing the electromagnetic field to $10^{-6}\lambda$ for ultra-high enhancement of field-matter interaction

Xiang-Dong Chen[1,2], En-Hui Wang[1,2], Long-Kun Shan[1,2], Ce Feng[1,2], Yu Zheng[1,2], Yang Dong[1,2], Guang-Can Guo[1,2] & Fang-Wen Sun [1,2✉]

Focusing electromagnetic field to enhance the interaction with matter has been promoting researches and applications of nano electronics and photonics. Usually, the evanescent-wave coupling is adopted in various nano structures and materials to confine the electromagnetic field into a subwavelength space. Here, based on the direct coupling with confined electron oscillations in a nanowire, we demonstrate a tight localization of microwave field down to $10^{-6}\lambda$. A hybrid nanowire-bowtie antenna is further designed to focus the free-space microwave to this deep-subwavelength space. Detected by the nitrogen vacancy center in diamond, the field intensity and microwave-spin interaction strength are enhanced by $2.0 \times 10^8$ and $1.4 \times 10^4$ times, respectively. Such a high concentration of microwave field will further promote integrated quantum information processing, sensing and microwave photonics in a nanoscale system.

---

[1] CAS Key Laboratory of Quantum Information, School of Physical Sciences, University of Science and Technology of China, Hefei 230026, People's Republic of China. [2] CAS Center For Excellence in Quantum Information and Quantum Physics, University of Science and Technology of China, Hefei 230026, People's Republic of China. ✉email: fwsun@ustc.edu.cn

Electromagnetic field can usually be focused at the scale of its wavelength. However, in pursuit of a strong interaction with matter, the manipulation of the electromagnetic field in a subwavelength space is one of the most important tasks in nanoscience researches and applications, ranging from integrated optics to biological sensing[1–4]. Nanostructures of dielectric[5–7], metallic[8–10], and two-dimensional materials[11,12] have been developed to tightly confine the electromagnetic field mainly based on the evanescent-wave coupling. For example, the plasmonic nanostructure has been used for the light field confinement at a scale smaller than $10^{-2}\lambda$[8–10]. These confinements can dramatically reduce the mode volume to highly increase the density of states and enhance the light−matter interaction at the nanoscale, which has harnessed the researches of single-molecule spectroscopy[13], nano laser[14], nonlinear optics[15], and solar energy[16].

Especially, the interaction between microwave field and matter at the nanoscale strongly drives the development of quantum information processing, sensing, and microwave photonics. The deep-subwavelength confinement of the microwave field will benefit the individual manipulation of multi-qubit[17,18]. Meanwhile, the enhancement of the local microwave field is of central importance to microwave-to-optics conversion[19,20], fast spin qubit manipulation[21], and hybrid quantum system coupling[22,23]. It indicates that the efficient localization and detection of microwave field at the nanoscale is highly required for developing a practical quantum information device. Furthermore, the wireless qubit manipulation with a compact and scalable system will decrease the power consumption and reduce the heat load in a cryostat[24,25]. However, directly pumping the qubit from far-field is usually inefficient[26]. And the gradient of the microwave is limited by the diffraction[27]. Though the in-plane slotted patch antenna has been demonstrated for the enhancement of local microwave field at the deep subwavelength scale[19,28], the Johnson noise of a large metal film will decrease the spin relaxation time[29,30], which is important for quantum computing and sensing.

Here, we study the field confinement based on the direct coupling between electromagnetic field and confined electrons in a low dimensional nano material. A tight confinement of a microwave field with an ultra-strong intensity is realized by utilizing the near field radiation of the electron oscillation in an Ag nanowire. Using the NV center in diamond as a noninvasive probe, we show that the microwave field can be localized down to 291 nm, corresponding to a scale of $10^{-6}\lambda$. For far-field spin manipulation, we design a hybrid nanowire-bowtie structure to focus the microwave field directly from the free space to a deep-subwavelength volume. As a result, the microwave-spin interaction strength is highly enhanced by observing a $1.4 \times 10^4$ times enhancement of the Rabi oscillation frequency, corresponding to increasing the field intensity by $2.0 \times 10^8$ times. Further considering the light guiding effect of the Ag nanowire, this antenna can be used for delivering and concentrating both light and microwave field. Subsequently, a wireless platform can be developed for the integrated quantum information processing and quantum sensing.

## Results

**The design of experiments**. As shown in Fig. 1a, the nanowire-bowtie hybrid antenna consists of an Ag nanowire with a diameter of 120 nm and a metallic bowtie structure. The gap between the two arms of the bowtie structure is $W_{\text{gap}} = 8\,\mu\text{m}$. The length of the bowtie structure is 6.5 cm, while the widths at the end and at the gap are 1 cm and 160 $\mu$m, respectively (details in Supplementary Fig. 1). A double-ridged horn antenna radiates

the microwave signal into the free space. The nanowire-bowtie structure then receives the far-field microwave (with a distance of approximate 20 cm).

The NV center in a single crystal diamond plate is generated by nitrogen ion implanting. The depth is approximate 20 nm. The ground state of NV center is a spin-triplet. The transition between the $m_s = 0$ and $m_s = \pm 1$ can be pumped by a resonant microwave. It subsequently changes the fluorescence intensity of NV center. To detect the microwave, we record the optically detected magnetic resonance (ODMR) of NV center under a continuous-wave microwave pumping. The contrast of the ODMR signal increases with the amplitude of the microwave field (Supplementary Fig. 2). To non-invasively map the localized microwave with a high spatial resolution, the charge state depletion (CSD) nanoscopy[31] is applied for the diffraction-unlimited ODMR measurement. It is based on the charge state manipulation and detection of NV center. The resolution of CSD nanoscopy is approximate 100 nm here, in comparison with the 500 nm resolution of the confocal microscopy (Supplementary Fig. 3). For the microwave field imaging in a large area, a wide field microscope is also used to detect the ODMR signal of NV center.

**Microwave localization and detection**. In Fig. 1b, we show the magnetic component of the microwave field near the Ag nanowire with a high spatial resolution. Here, without an external magnetic field, the resonant microwave frequency for NV center spin transition is 2.87 GHz. The result shows that the near-field microwave is confined near the Ag nanowire. The width of the cross-section profile is $291 \pm 10$ nm, corresponding to $2.8 \times 10^{-6}\lambda$, where $\lambda \approx 10.4$ cm is the microwave wavelength in vacuum. The distribution of the magnetic component (the insert of Fig. 1b) can be well fitted by a reciprocal function:

$$|B_{\text{MW}}(r)| \propto \frac{1}{\sqrt{r^2 + r_0^2}}, \tag{1}$$

where $r$ is the distance from the nanowire in the $xy$ plane and $r_0$ is determined by the radius of the nanowire and the depth of NV center. The fitting result shows that $r_0 = 84 \pm 3$ nm, which matches the expectation. In contrast, the evanescent-field coupling shows an exponential decay as a function of distance[4].

To reveal the mechanism of the microwave confinement, we separate the ODMR signal from the four categories of NV centers with different symmetry axes and obtain the vector information of the localized microwave field, as shown in Fig. 2a. Here, the $x$, $y$, and $z$ axes are defined as the edges of the diamond plate. The symmetry axes of the four categories of NV centers are shown as NV1 ($-\sqrt{2}$, 0, 1), NV2 ($\sqrt{2}$, 0, 1), NV3 (0, $\sqrt{2}$, 1), NV4 (0, $-\sqrt{2}$, 1). An external static magnetic field $\mathbf{B_0}$ is applied to split the four categories (NVi) according to Zeeman effect, which is shown in Fig. 2b. The difference in resonant frequencies is not large here. No obvious wavelength-selectivity of microwave localization is observed in this experiment. Then, we can assume that the localized microwave field distribution with different resonant frequencies is the same. By recording the ODMR signal of NVi centers at different positions, we obtain the distribution of the magnetic projection $B_{\text{MWi}}$, which is perpendicular to the NVi centers' symmetry axis.

To precisely map the microwave vector distribution, the position of the nanowire is firstly located according to the fluorescence enhancement of NV center, as shown in Fig. 2c. We find that the position of the magnetic component's maximum does not always match the position of the nanowire. As shown in Fig. 2d, the maximum of $B_{\text{MW2}}$ is approximate 100 nm away from the nanowire. In Fig. 2e, f, by simultaneously detecting the fluorescence signals with different microwaves pumping,

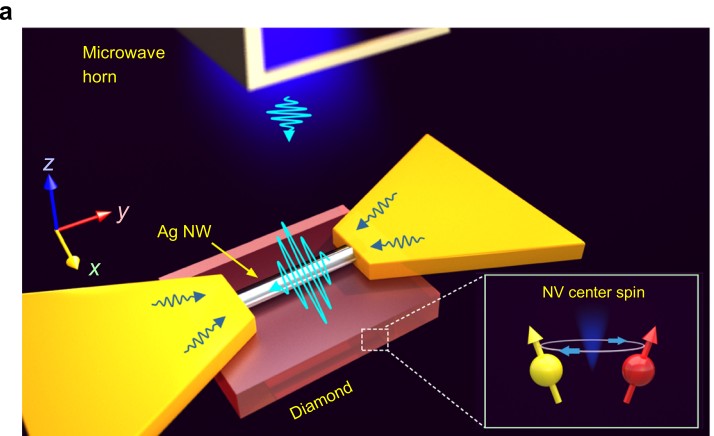

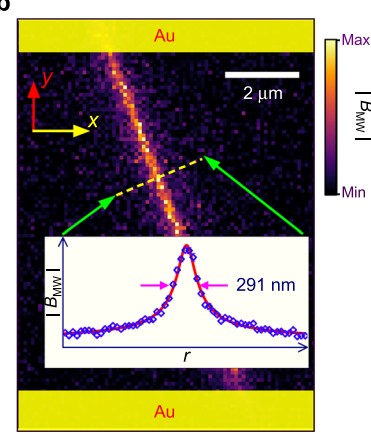

**Fig. 1 The principle of localizing and detecting the microwave field. a** Sketch of the nanowire-bowtie antenna. A single-crystal diamond plate is placed under the nanowire. The spin state transition of NV center in diamond is pumped by the localized microwave. **b** The image of the microwave distribution is obtained by recording the spin-state transition of NV centers at different positions with CSD nanoscopy. The insert is the integrated cross-section profile. The solid line is the fit with Eq. (1). The error bars represent the standard error. The power of the microwave that is radiated by the horn antenna is 14 $\mu$W.

we further highlight the difference of the distributions with $|B_{MW2}| - |B_{MW1}|$ and $|B_{MW4}| - |B_{MW3}|$, respectively. With the structure in Fig. 2, the amplitudes of $B_{MW3}$ and $B_{MW4}$ are almost the same, while the amplitude of $B_{MW1}$ is mirroring $B_{MW2}$ with respect to the $yz$ plane. The results indicate that the vector of the magnetic component is in the $xz$ plane. Comparing with the simulation (Supplementary Note 5), it confirms that the magnetic component of the local microwave follows the distribution of the magnetic field around a straight line current that is transmitting through the Ag nanowire. The small mismatch between the simulation and the experiments might be caused by the error of nanowire's location, the distribution of NV center in $z$-axis, and the accuracy of CSD nanoscopy resolution's estimation.

Based on the results of Figs. 1 and 2, we deduce that the free-space microwave is collected by the bowtie structure, then the oscillating currents transmit through the Ag nanowire and generate a strong localized microwave field. Due to the small size of nanowire, the localized microwave is highly confined according to the Biot-Savart law. The localization of microwave is mainly determined by the Ag nanowire. The distribution of localized microwave field will be the same with different wavelengths. In addition, since the bowtie structure is a typical wide band antenna, the factor of localized microwave enhancement varies very small with different microwave frequencies in this experiment.

**Far-field spin manipulation**. The tight confinement leads to a significant enhancement of the localized microwave field's intensity. It subsequently enhances the interaction with a spin qubit. In Fig. 3, we compare the localized microwave field with three different structures: the nanowire-bowtie hybrid antenna, the bowtie antenna without Ag nanowire, and no antenna. The results show that, with a nanowire-bowtie antenna, the microwave is significantly enhanced near the Ag nanowire, while a bowtie antenna without Ag nanowire slightly increases the localized microwave field in the gap between the two arms.

The microwave field enhancement can be used for fast and high spatial resolution spin qubit manipulation. Here, we use it to pump the Rabi oscillation of NV center. As shown in Fig. 3d, with a nanowire-bowtie antenna, the Rabi frequency of the NV center under the Ag nanowire is approximate 1.6 $\mu s^{-1}$ with a 14 $\mu$W microwave excitation. Without Ag nanowire, the Rabi oscillation frequency of the NV center in the gap of the bowtie antenna is approximate 0.89 $\mu s^{-1}$ with a 21 W microwave

excitation. In contrast, without any nanostructure, the Rabi oscillation frequency is only 0.14 $\mu s^{-1}$ under a 21 W microwave excitation. The results indicate that, by utilizing the nanowire-bowtie antenna for spin manipulation, the Rabi frequency can be improved by at least $1.4 \times 10^4$ times, corresponding to increasing the local microwave intensity by $2.0 \times 10^8$ times. The observation of Rabi oscillation also indicates that the coherence of both the spin and the localized microwave is preserved, which is crucial for quantum applications. Note that, single NV centers are used to measure the slow Rabi oscillation in Fig. 3e, f. The amplitude of Rabi oscillation with NV center ensemble in Fig. 3d is smaller than that with a single NV center in Fig. 3e, f. It is because there are four possible symmetry axes with NV center ensemble, and we only measure the Rabi oscillation of NV center with one particular axis. In addition, the inhomogeneous broadening of NV center ensemble also decreases the visibility of Rabi oscillation.

The individual addressing of multi-qubit from far-field can be further explored by utilizing the polarization dependence of the localized microwave enhancement. Here, we rotate the horn antenna to change the polarization of the free-space microwave. The results in Fig. 4 show that, with an electrical polarization parallel to the bowtie-nanowire antenna, a stronger localized microwave field is observed near the nanowire-bowtie antenna. The polarization isolation of the localized microwave intensity with the nanowire-bowtie antenna is higher than 20 dB, but lower than 40 dB. Therefore, encoding the microwave for different spin manipulation into the polarization, the nanowire-bowtie antennas with different directions can be used to selectively manipulate the qubits at different positions from the far-field.

## Discussion

Integrating and miniaturizing the electrical and optical device is essential for the practical quantum processing and sensing applications[32,33]. Various electrical conductive and ferromagnetic structures have been studied to transmit the microwave signal for spin manipulation at the nanoscale[34,35]. Our method provides a solution to efficient spin manipulation from the far-field. The Johnson noise from an Ag nanowire is negligible in comparison with the metal film (Supplementary Fig. 4). It will simplify the quantum processing device, and avoid the thermal leakage in a cryostat. In addition, the light-guiding effect of an Ag nanowire can be utilized to optically pump and collect the fluorescence of individual qubit[36–38]. The simultaneous integration of electrical

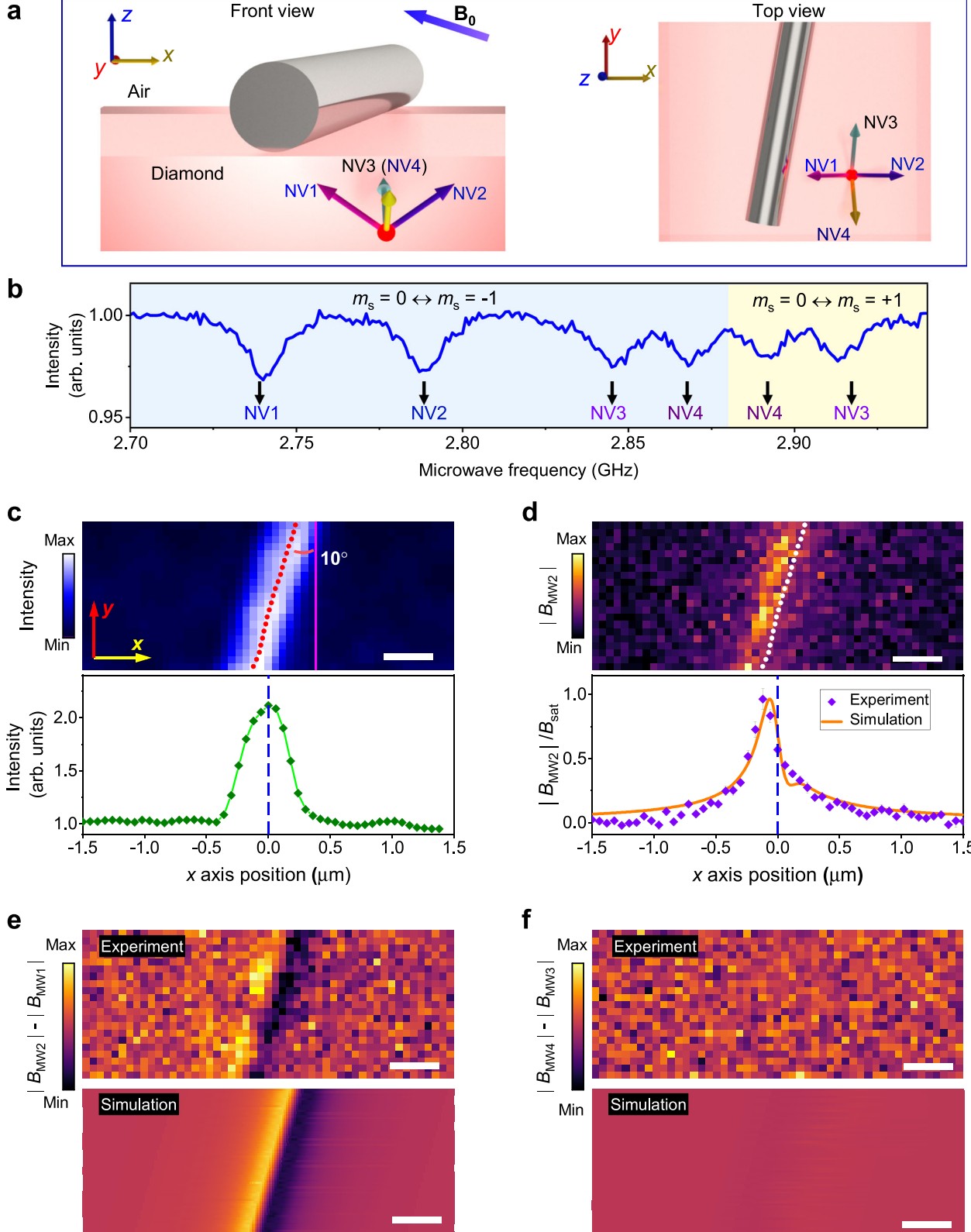

and optical components certainly will help to develop a compact and integrated quantum processing device.

Focusing and detecting the nanoscale electromagnetic field can be further used to enhance the sensitivity of the spin-based metrology. Recently, the magnetic concentrator of ferrite material has been applied for the detection of the static magnetic field with NV center[39,40]. The key is how small the electromagnetic

field can be focused and detected. Utilizing the nanowire-bowtie antenna, we can improve the sensitivity by $1.4 \times 10^4$ times. Combining with the coherent spin manipulation[41], it will help to realize the ultra-weak microwave signal sensing, such as for a quantum radar. The enhanced fluorescence intensity of NV center with Ag nanowire will also help to improve the sensitivity of sensing. However, the oxidation of Ag nanowire in the

**Fig. 2 Detection of the microwave field vector at the nanoscale. a** The illustration of the direction of nanowire and NV center axes. **b** Frequency-scanning ODMR results of NV centers with four symmetry axes. **c** The fluorescence intensity of NV centers is enhanced due to the interaction with the Ag nanowire. The cross-section profile in the bottom is used to locate the relative position of nanowire (red points in the image). **d** The image of $|B_{MW2}|$. The cross-section profile is the integrated signal of the whole image. The solid line is the simulation with a straight line current. The value is normalized by the saturation amplitude $B_{sat}$, as defined in Supplementary Eq. (2). Error bars represent the standard error. **e, f** The microwave vector is revealed by comparing different projections. The result in **e** records the ratio of fluorescence intensity with $B_{MW1}$ to the fluorescence with $B_{MW2}$ and **f** is recorded as the ratio of fluorescence with $B_{MW3}$ to fluorescence with $B_{MW4}$. With a weak microwave pumping, the images of (**e**) and (**f**) approximate to the distribution of $|B_{MW2}| - |B_{MW1}|$ and $|B_{MW4}| - |B_{MW3}|$, respectively (see Supplementary Note 2). The scale bars in all the images are 400 nm in length. And the power of the microwave source is set to 174 $\mu$W.

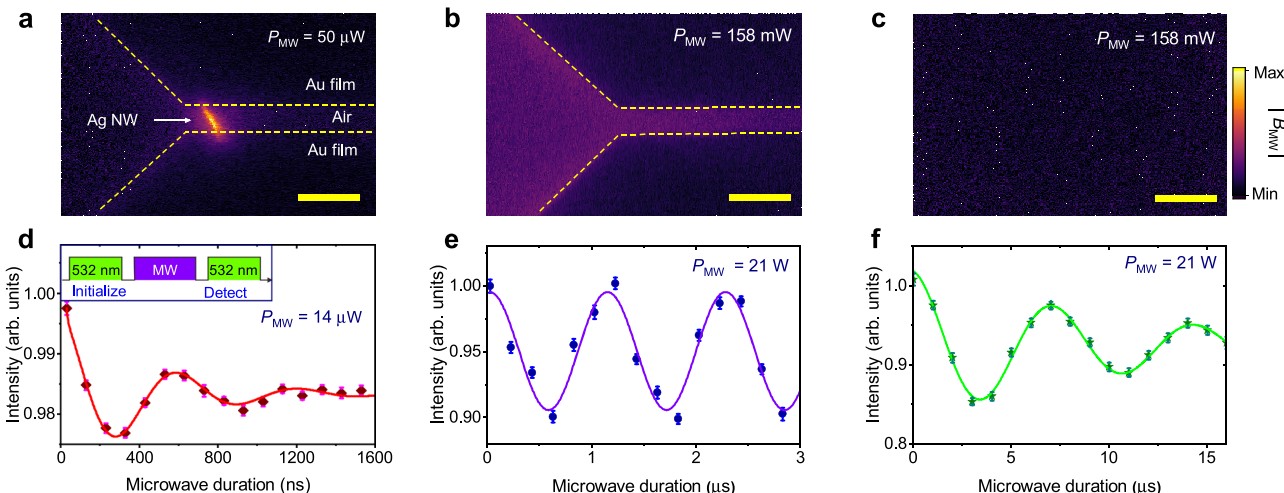

**Fig. 3 Spin manipulation with the localized microwave field.** The wide-field images of the microwave field enhancement with different structures: **a** nanowire-bowtie antenna; **b** bowtie antenna without nanowire; **c** no structure on the diamond surface. The scale bars are 20 $\mu$m in length. **d–f** Rabi oscillations of NV center with different structures in (**a–c**), respectively. The inset in **d** shows pulse sequences for the Rabi oscillation measurement. The Rabi oscillation in **d** is measured with NV center ensemble under the nanowire. And the results in **e** and **f** are measured with single NV centers. $P_{MW}$ is the power of the microwave source. Error bars represent the standard error.

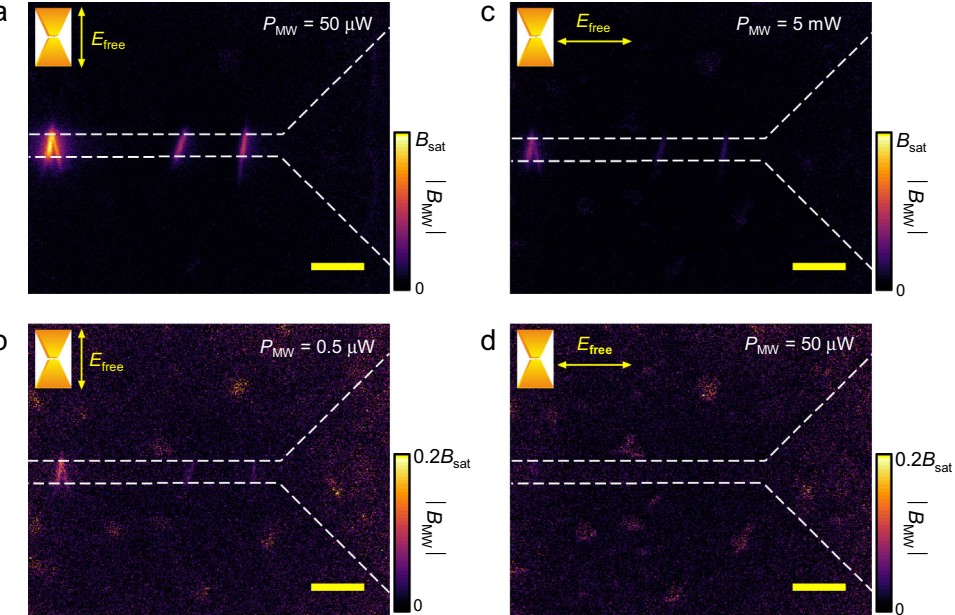

**Fig. 4 The localized microwave field distribution is changed with the polarization of the free-space microwave. a, b** The electrical polarization of the free-space microwave is parallel to the direction of nanowire-bowtie structure. **c, d** The electrical polarization is perpendicular to the nanowire-bowtie structure. The results are obtained with the wide field microscopy. The scale bars are 20 $\mu$m in length. $E_{free}$ denotes the electrical component of the free-space microwave.

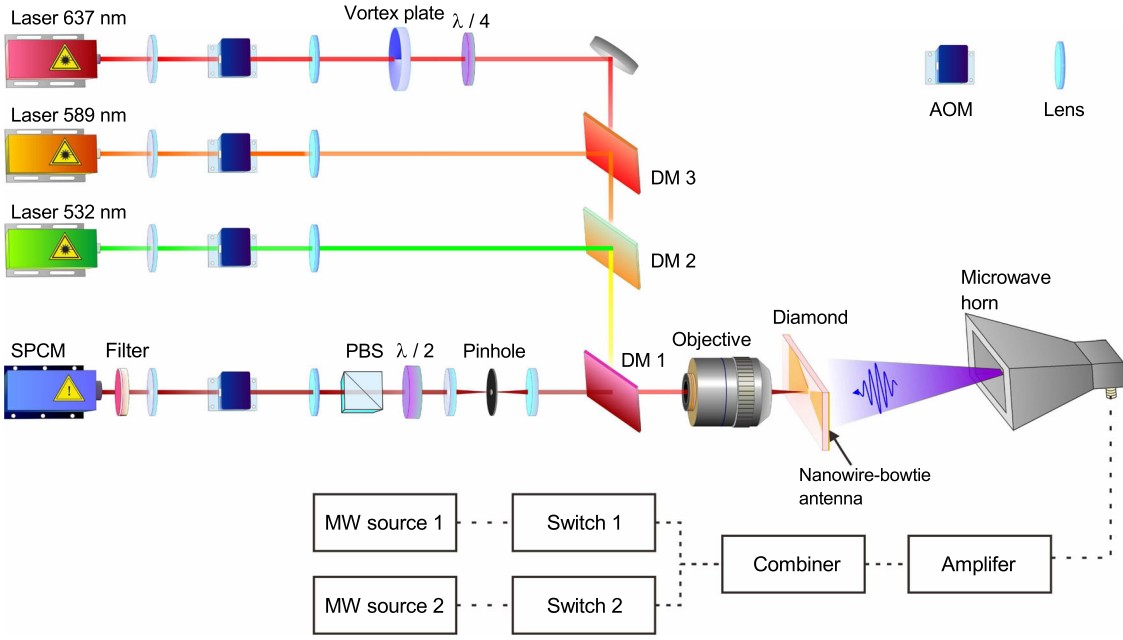

**Fig. 5 The schematic diagram of the experimental setup for the CSD nanoscopy.** DM1-3, long-pass dichroic mirror with edge wavelengths of 658.8, 536.8, and 605 nm, respectively; AOM acousto-optic modulator; SPCM single-photon-counting modulator; PBS polarizing beam splitter.

atmosphere will change the conductivity of nanowire networks. The potential solutions include using the monolayer $SnO_2$ protected nanowire[42].

In conclusion, we have demonstrated the high concentration of a microwave field by utilizing the confined electron's oscillation in a low dimensional material. The results can be used for integrated quantum information processing and high-sensitivity quantum sensing.

## Methods

**Sample preparation**. The electrical grade diamond plates with {100} surface and <110> edges are purchased from Element 6. The size of the plate is $2 \times 2 \times 0.5$ mm$^3$. The NV center ensemble is produced through nitrogen ion implanting with an energy of 15 keV and a dosage of $10^{13}$/cm$^2$. The diamond is annealed at 850 °C for 2 h to improve the production efficiency of NV centers. The density of NV centers is estimated to be approximate 5000/$\mu$m$^2$. The high-density NV center samples are used to image the distribution of the local microwave. A low-density NV center sample has also been used to measure the Rabi oscillation in Fig. 3e,f. It is produced by nitrogen ion implanting with a dosage of $10^9$/cm$^2$. After the NV center is produced, Ag nanowires are dropped on the surface of the diamond with a spin processor. Then, a small metallic bowtie structure of chromium/gold (5/200 nm thickness) film is produced on the diamond surface through lift-off. Finally, a large in-plane bowtie antenna is made with a copper foil tap. The Au film on the diamond plate is ohm connected to the copper tape with silver glue.

**Experimental setup**. The CSD nanoscopy setup for ODMR measurement is based on a home-built confocal microscope, as shown in Fig. 5. The diamond plate is mounted on a piezo-stage (P-733.3DD, PI). CW lasers with wavelengths of 532 (Coherent), 589 (MGL-III-589nm, New Industries Optoelectronics), and 637 nm (MRL-III-637nm, New Industries Optoelectronics) are modulated by acousto-optic modulators (AOMs, MT200-0.5-VIS, AA). A vortex phase plate (VPP-1a, RPC photonics) is used to produce a doughnut-shaped 637 nm laser beam. The lasers pump the NV center in diamond from the backside through an objective (Leica) with 0.7 NA. The collected fluorescence is time-gated by another AOM. Then, it is detected by a single-photon-counting-module (SPCM-AQRH-15-FC, Excelitas) after passing through a long-pass filter (edge wavelength 668.9 nm, Semrock). In the wide-field microscope, a 532 nm CW laser (MLL-III-532nm, New Industries Optoelectronics) is used to pump the NV center. The fluorescence is detected by a CCD camera (iXon897, Andor).

Two microwave generators (SMB 100A and SMA 100A, Rhode&Schwartz) are used to produce microwave signal with different frequencies. The microwave pulse is controlled by microwave switches (ZASWA-2-50DR, MiniCircuits). Then, the two channels are combined by a combiner (ZFRSC-42-S, MiniCircuits) and amplified by a microwave amplifier (60S1G4A, Amplifier Research). The microwave is radiated into free space by a horn antenna (LB-2080-SF, Chengdu AINFO Inc.).

## Data availability

The data that support the findings of this study are available from the corresponding author upon reasonable request.

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

## Acknowledgements

This work was supported by the National Key Research and Development Program of China (No. 2017YFA0304504); Anhui Initiative in Quantum Information Technologies (AHY130100); National Natural Science Foundation of China (Nos. 91536219, 91850102). The sample preparation was partially conducted at the USTC Center for Micro and Nanoscale Research and Fabrication.

## Author contributions

X.C. and F.S. conceived the idea. X.C. performed the experiments. E.W. and C.F. prepared the samples. L.S. and Y.Z. built the electrical setup. X.C., Y.D., and F.S. analyzed the data. All authors contributed to the discussion and editing of the manuscript.

## Competing interests

The authors declare no competing interests.
