## [Peer Review File · Nature Communications]

REVIEWER COMMENTS

Reviewer #1 (Remarks to the Author):

The manuscript "Focus the electromagnetic field to $10^{-6}\lambda$ for ultra-high enhancement of field-matter interaction" by Xiang-Dong Chen et al. reports an extreme localization of microwave field down to $10^{-6}\lambda$ based on the direct coupling with confined electron oscillations in a nanowire. With the hybrid nanowire bowtie antenna, the free-space microwave is focused to deep-subwavelength space. And the field intensity and microwave-spin interaction strength are detected by the nitrogen vacancy center in diamond. The results are supported by the provided data, however mechanism of the localization of microwave field with Ag nanowire is not fully explained. Therefore, a major revise is needed before publication.

1. What is the specific wavelength of the microwave, is there wavelength selectivity of the nanowire-bowtie antenna?
2. Why silver nanowires were chosen to focus microwaves? Will the oxidation of silver nanowires have an effect on the localization of microwave field?
3. Can silver nanowires be replaced with silicon materials?
4. A brief statement should be given about the research status and problems of microwave focusing.

Reviewer #2 (Remarks to the Author):

The authors present a study of em-field concentration in a nano-antenna situation tested with NV centers. The subject of the study is interesting and timely since spin manipulation of NV centers is important for the rapidly developing field of (low power) quantum information science.

I would like to have the following issues clarified:

Major:

I do not understand Fig. 3. Why is the Rabi oscillation of the NV center(s) with the nanowire antenna almost non-existent (and therefore UNSuitable for quantum information processing) while the naked antenna exhibits nice oscillations. How can Fig. 3e be due to a single NV center, what is the proof for that?

Minor:

To what data and how was eq. (1) fitted? Does that fit to the expectation from the implantation parameters?

We thank the Editor and Reviewers for their helpful comments! We modified the manuscript according to those comments. The changes of the manuscript are marked by blue color. And main changes are summarized as:

- 1, The mechanism of microwave localization was further explained in the third paragraph of page 4.
- 2, The wavelength of microwave was clarified. And the wavelength dependence was discussed.
- 3, The reason of using Ag nanowire was explained. And the oxidation effect was discussed in the discussion section.
- 4, The introduction of the nanoscale microwave field modulation for spin manipulation was added in the introduction section.
- 5, The difference between the Rabi oscillations with high density NV center ensemble and single NV center was explained in the first paragraph of page 5.
- 6, The fitting with Eq. (1) was clarified.
- 7, We optimized the color schemes of Figures to ensure that red and green colors were not simultaneously used, meeting the requirements of the journal.
- 8, In Figs. 2e and f, by simultaneously measuring the fluorescence with different microwave pumping, we presented the results of I_{MW1}/I_{MW2} and I_{MW3}/I_{MW4} , respectively. In previous manuscript, we incorrectly wrote that they were $|B_{MW2}|/|B_{MW1}|$ and $|B_{MW4}|/|B_{MW3}|$. It was a typing error. They actually approximated to the distributions of $|B_{MW2}| - |B_{MW1}|$ and $|B_{MW4}| - |B_{MW3}|$. The mistake was corrected and explained in the revised manuscript.
- 9, Grammar errors were checked.
- 10, Format of the manuscript was further adjusted according to the requirement of journal.

In the following, we present the point-by-point response to the comments from the Reviewers.

Reviewer 1

Comment 1

“The manuscript “Focus the electromagnetic field to $10-6\lambda$ for ultra-high enhancement of field-matter interaction” by Xiang-Dong Chen et al. reports an extreme localization of microwave field down to $10-6\lambda$ based on the direct coupling with confined electron oscillations in a nanowire. With the hybrid nanowire bowtie antenna, the free-space microwave is focused to deep-subwavelength space. And the field intensity and microwave-spin interaction strength are detected by the nitrogen vacancy center in diamond. The results are supported by the provided data, however mechanism of the localization of microwave field with Ag nanowire is not fully explained. Therefore, a major revise is needed before publication.”

Our reply:

We thank the Reviewer for the positive and useful comments! We carefully modified the manuscript according to these suggestions.

According to the experimental results, the magnetic vector of the localized microwave field follows the distribution of magnetic field around a straight line current, which is transmitting through the Ag nanowire. Meanwhile, the factor of the localized microwave enhancement is changed by changing the size of bowtie structure, as in Supplementary Fig. 1c. We deduce that the localized microwave is generated as follows: the bowtie structure receives the free-space microwave; then, the oscillating current transmits through the nanowire and generates a localized microwave field. Due to the small size of nanowire (a small r_0 in Eq. (1)), the microwave is highly confined and enhanced.

Figure R1.1 The microwave localization with a small nanowire-bowtie antenna. (a) The localization of microwave near the nanowire is observed with the wide field microscopy. The insert shows the image of the antenna on the surface of a diamond plate. (b) The Rabi oscillation near the nanowire.

We have tested a nanowire-bowtie antenna with a very small bowtie structure (2 mm in length and 2 mm in width), as shown in Figure R1.1. The extreme microwave localization has also been observed near the nanowire, though the localized microwave enhancement was lower than that in the main text. It confirms that the localization of microwave field is mainly determined by the nanowire. And the nanowire-bowtie structure can be used for the localization of microwave with different frequencies.

In the revised manuscript, we fully explained the mechanism of microwave localization in the third paragraph of page 4.

Comment 2 “*What is the specific wavelength of the microwave, is there wavelength selectivity of the nanowire-bowtie antenna?*”

Our reply:

In the manuscript, we showed the confinement of microwave with the frequency ranging from 2.74 to 2.87 GHz. It corresponded to the wavelength of 10.9 - 10.4 cm. Extreme microwave localization has been observed with all these wavelengths.

Figure R1.2 The Rabi oscillation was measured with different microwave frequencies. (a) The shape of antenna is revealed in the fluorescence image. (b) The Rabi oscillation results.

As explained in the reply of comment 1, the localization of microwave is determined by the nanowire, while the factor of microwave enhancement is affected by the bowtie structure. Typically, a bowtie structure is used as a wideband antenna. Since the change of wavelength was small in the manuscript, we did not observe a significant wavelength selectivity. As shown in Supplementary Fig. 1c, the microwave field enhancement changes only 5 dB when the size of antenna is changed from 3 cm to 6 cm. In addition, we have measured the Rabi oscillation with different microwave frequencies. As shown

in Figure R1.2, the Rabi oscillation frequency (amplitude of local microwave field) only changes 16% when the microwave frequency changes 124 MHz. It is negligible comparing with the local field enhancement.

In practical applications, the bowtie structure can be easily modified according to the wavelength. And we can further decrease the wavelength dependence by optimizing the designing of bowtie antenna.

In the revised manuscript, we clearly described the wavelength of microwave and discussed how the wavelength might affect the results.

Comment 3 “*Why silver nanowires were chosen to focus microwaves? Will the oxidation of silver nanowires have an effect on the localization of microwave field?*”

Our reply:

In this experiment, the silver nanowires were chosen because they enhanced both the fluorescence intensity of NV center and the local microwave field. The enhancement of fluorescence intensity can improve the signal to noise ratio of NV center spin detection. And the light guiding effect of the nanowire can be further explored to pump and collect the spin-dependent fluorescence of NV center.

Oxidation is a problem that we have encountered in the experiment. The electrical conductivity of nanowire networks may change within few days or weeks due to the random oxidation, as shown in Figure R1.3. There are also exceptions. We tested a sample that did not show any changes of conduction in more than one month. Currently, we are testing the nanowires with monolayer SnO₂ on the surface to protect them from oxidation (J. Am. Chem. Soc. 141, 13977-13986(2019)) for further applications. We hope it can solve the problem. Other methods, such as encapsulating the structure with hBN film, are also in consideration.

Figure R1.3 The conductivity of nanowire networks was changed due to oxidation. The local microwave distribution was measured with a wide field microscope here.

In the second paragraph of discussion section, we added the discussion of oxidation, and explained why the silver nanowires were used here.

Comment 4 “*Can silver nanowires be replaced with silicon materials?*”

Our reply:

Silicon materials are widely used in today’s integrated electronics and photonics devices. One important advantage of utilizing silicon materials and other CMOS-compatible materials is that they are convenient for the large-scale production.

With proper ion doping, the electrical conductivity of silicon can be significantly improved. However, it is still much lower than that of silver. For the design in this work, the localized microwave field is mainly determined by the current in Ag nanowire. High conductivity of the nanowire is important for the localization of microwave. Replacing the silver nanowire with silicon nanowire may affect the localization of microwave field. In addition, the silicon is not transparent for light with the wavelength of 600-800 nm (NV center fluorescence). The local fluorescence intensity of NV center will also be changed by replacing the nanowire with silicon materials. Therefore, to apply the silicon materials in the experiment, the design should be modified, and more experimental tests are needed.

Comment 5 “*A brief statement should be given about the research status and problems of microwave focusing.*”

Our reply:

Thanks for the suggestion! In the second paragraph of introduction section, we added more introductions to the nanoscale microwave field modulation for spin manipulation.

Microwave pulses are widely used for the manipulation of quantum systems, such as spin in solids, trapped ion, and superconducting qubit. Coaxial cables are used to transmit the microwave for the near-field manipulation of a qubit. And a microwave horn antenna can be used to pump the qubit from far field. The far-field manipulation is much easier, especially for the quantum system in a cryostat. However, the gradient and strength of the free-space microwave are limited by the diffraction. In the studies of microwave photonics, the coplanar antenna with a small size slot (~ 100 nm) has been used to directly enhance the local electromagnetic field. However, the Johnson noise of the metal film will decrease the relaxation time of a nearby spin qubit. Therefore, the in-plane slotted patch antenna is not suitable for the manipulation of a qubit. We need to develop a new structure for efficient and selective qubit manipulation.

Reviewer 2

Comment 1

“The authors present a study of em-field concentration in a nano-antenna situation tested with NV centers. The subject of the study is interesting and timely since spin manipulation of NV centers is important for the rapidly developing field of (low power) quantum information science.”

Our reply:

We appreciate the positive comments from the Reviewer. In this revised manuscript, we modified the presentation of manuscript according to the questions from the Reviewer.

Comment 2

“I do not understand Fig. 3. Why is the Rabi oscillation of the NV center(s) with the nanowire antenna almost non-existent (and therefore UNSuitable for quantum information processing) while the naked antenna exhibits nice oscillations. How can Fig. 3e be due to a single NV center, what is the proof for that?”

Our reply:

Thanks for the careful reading!

Samples with high density NV ensemble and low density NV (single NV centers) are both used in the experiment. Usually, the contrast of spin resonance signal with high density NV center ensemble is much lower than that with a single NV center. However, the using of nanowire-bowtie antenna is not the reason of low Rabi oscillation contrast with NV center ensemble. There are two main reasons that cause a low contrast of Rabi oscillation with NV center ensemble. First, there are four possible axes of NV center ensemble. If we only measure the Rabi oscillation signal of NV center with one particular axis, the fluorescence of other axes will contribute to the background. Second, the inhomogeneous broadening decreases the contrast. And the high density defects near NV centers significantly decrease the coherence time of NV center. Therefore, we observe that the contrast of Rabi oscillation with NV center ensemble is lower than a single NV center. And the decay of the Rabi oscillation with NV center ensemble is much faster than that with a single NV. Same results are observed with other type's antenna. This problem can be solved by optimizing the sample preparation process, such as using perfectly aligned NV centers.

In Fig. 3d, we demonstrated the enhancement of localized microwave with a nanowire-bowtie structure by recording the Rabi oscillation of NV center ensemble. It is because we need to measure the signal of NV center just under the nanowire. Using a diamond with high density NV centers is much easier to find NV centers near the nanowire than using low density NV centers. In contrast, the Rabi oscillation of a single NV center is measured in Fig. 3e (bowtie antenna without nanowire) and 3f (no

structure on diamond). It is because the microwave without nanowire is much weaker than that with nanowire-bowtie antenna. The period of Rabi oscillation without nanowire would be much longer than that with nanowire-bowtie antenna. And it will be longer than the dephasing time of NV center ensemble. Therefore, we need to use a single NV to measure the Rabi oscillation with the weak microwave in a relatively large area.

In our experiment, the single NV center is imaged with the confocal microscopy, as shown in Figure R2.1. The single NV center can be further determined by measuring the second-order correlation function. In future applications, to place a single NV center near the nanowire, we can move the nanowire with an AFM probe, or produce single NV centers near the nanowire through laser writing technique.

In the first paragraph of page 5, we explained the Rabi oscillation contrast with NV center ensemble. We also clearly stated that diamond samples with high density NV and low density NV were both used in the experiment.

Figure R2.1 The measurement with a single NV center. (a) Confocal image of the low density NV centers. The single NV centers can be distinguished in the image as separated points. The insert shows a typical $g^{(2)}$ function measurement with a single NV center. (b) The Rabi oscillation of a single NV center. The NV center is directly pumped by the microwave that is radiated by a horn antenna.

Comment 3

“To what data and how was eq. (1) fitted? Does that fit to the expectation from the implantation parameters?”

Our reply:

In the insert of Fig. 1(b), to estimate the width of local microwave distribution, we fitted the experimental data (the position dependent magnetic amplitude crossing the nanowire) with Eq. (1). Specifically, the normalized signal is fitted as $|B_{MW}(r)| =$

$\frac{1}{\sqrt{r^2+r_0^2}}$. r is the relative position in xy plane, where we set the position of nanowire to

$r = 0$. The parameter r_0 is determined by the radius of nanowire (r_{NW}) and the depth of NV center (d_{NV}). The fitting gives the parameter as $r_0 = 84 \pm 3 \text{ nm}$. In our experiment, the average radius of nanowire is 60 nm. And the simulation with SRIM software gives an average depth of NV center of 20 nm.

For the magnetic field distribution with a straight line current, we can write $r_0 = r_{NW} + d_{NV}$. So, we would say it roughly matches the expectation. Some other factors, such as the resolution of microscopy, should be considered for a more accurate estimation. To confirm whether the local microwave is generated by a confined straight line current, we still need to perform the microwave vector detection, as in Figure 2.

In the third paragraph of page 2, we clarified the fitting with Eq. (1).

REVIEWERS' COMMENTS

Reviewer #1 (Remarks to the Author):

In the previous version, the authors presented the excellent properties of the hybrid nanowire-bowtie antennas to focus the free-space microwave to the subwavelength region. The work is novelty and interesting, while the relevant physical mechanisms and experimental details are not fully discussed. So I am moderately positive about this paper. In this revised manuscript, the authors have made appropriate modifications to the manuscript and answered my questions. In general, the hybrid antennas do have their own unique advantages, and this work is indeed of good quality. So I believe it is suitable and worth publishing in Nature Communications.

Reviewer #2 (Remarks to the Author):

I find that the authors have answered my questions satisfactorily; also the questions of the other referee were interesting and I find them also answered. The manuscript is clear now. In my view it can be published as is.

Reviewer 1

Comment 1

“In the previous version, the authors presented the excellent properties of the hybrid nanowire-bowtie antennas to focus the free-space microwave to the subwavelength region. The work is novelty and interesting, while the relevant physical mechanisms and experimental details are not fully discussed. So I am moderately positive about this paper. In this revised manuscript, the authors have made appropriate modifications to the manuscript and answered my questions. In general, the hybrid antennas do have their own unique advantages, and this work is indeed of good quality. So I believe it is suitable and worth publishing in Nature Communications.”

Our reply:

We thank the Reviewer for her/his professional comments during the peer review process!

Reviewer 2

Comment 1

“I find that the authors have answered my questions satisfactorily; also the questions of the other referee were interesting and I find them also answered. The manuscript is clear now. In my view it can be published as is.”

Our reply:

We thank the Reviewer for her/his professional comments during the peer review process!